# Magnetic resonance angiography in diagnostic long-term follow-up of primary patency of the MOTIV drug-eluting bioresorbable vascular scaffold in the region below the knee: 5 years of experience

Christian Nasel [1,2,3]*, Mario Kirschner[1], Karoline Rizzi[1], Nicola Schweinhammer[4], Ewald Moser[3]

1 Department of Radiology, University Hospital Tulln, Karl Landsteiner University of Health Sciences, Tulln, Austria, 2 Department of Medical Imaging and Image-guided Therapy, Medical University of Vienna, Vienna, Austria, 3 Center for Medical Physics and Biomedical Engineering, Medical University of Vienna, Vienna, Austria, 4 Department of Neurology—University Hospital Tulln, Karl Landsteiner University of Health Sciences, Tulln, Austria

☯ These authors contributed equally to this work.
* christian.nasel@meduniwien.ac.at

## Abstract

### Purpose

Treatment of peripheral artery disease (PAD) in the region below the knee (BTK) is dissatisfying as failure of treated target lesions (TLF) is frequent and diagnostic imaging is often challenging. In the BTK-region metallic drug-eluting stents (mDES) yielded best results concerning primary patency (PP), but also annihilate signal in magnetic resonance angiography (MR-A). A recently introduced non-metallic drug eluting bioresorbable Tyrocore® vascular scaffold (deBVS), that offers an option for re-treatment of lesions due to its full degradation within 3–4 years after placement, was investigated with respect to its compatibility with MR-A to unimpededly depict previously treated target lesions.

### Methods

Patency of the deBVS in the BTK-region was assessed retrospectively using contrast enhanced MR-A of the lower limbs in patients with PAD of Lafontaine-grades II-IV (n = 19). Clinically driven MR-A censoring was triggered by an assumed target lesion failure (CD-TLF), which served to compute the probability of PP during the observation period of 5 years. Compatibility of this particular deBVS with MRI was additionally proven via in-vitro experiments.

### Results

The scaffold was found to be fully compatible with MRI. The normalised intra-luminal signal measured in MR-A increased significantly after successful deBVS-placement. The

**Data Availability Statement:** All relevant data are within the manuscript and its Supporting Information files.

**Funding:** This manuscript was supported by D&S Biotrade, who are willing to cover publication fees. This support was paid directly to PLoS. No additional external funding was received for this study. The funders had no role in study design, data collection and analysis, decision to publish, or preparation of the manuscript.

**Competing interests:** The authors have declared that no competing interests exist

retrospective 5-years PP-probability was 0.87 (CI95%: [0.71,1.0]) with 2 stent-occlusions observed after 90 days. No major adverse events occurred.

## Conclusion

Assessment of PAD in the BTK-region after placement of the Tyrocore®-deBVS using MRA is feasible. The promising high PP-probability after 5-years and the persistent full interpretability of treated target lesions by MR-A after stent-placement encourage further prospective assessment of this deBVS in treatment of PAD in the BTK-region.

## Introduction

Treatment of peripheral artery disease (PAD) in the region below the knee (BTK) is still dissatisfying due to frequent occurrence of target lesion failures (TLF). In the BTK-region the use of metallic drug eluting stents (mDES) proved superior to treatment of PAD with bare metal stents or percutaneous transluminal angioplasty (PTA) only [1–4]. Though the management of this disease is primarily driven by clinical symptoms, imaging with direct angiographic assessment of the vessel state employing various modalities remains an essential part of therapy planning in PAD. Generally, calcifications and small vessel diameters limit the use of Doppler ultrasound (DUS) and computed tomography angiography (CT-A), but imaging is even more restricted, if metallic implants, like a mDES, are deployed in lesions during an endovascular recanalisation. Metallic implants especially affect magnetic resonance imaging (MRI) and MR-angiography (MR-A) by distortion and annihilation of the regional vessel signal, though otherwise MRI is most robust, e.g., in case of heavy vessel calcifications.

The advent of a new type of drug eluting bioresorbable vascular scaffolds (deBVS) opened new possibilities of diagnostic imaging and subsequent treatment of coronary artery disease and of PAD [5,6]. Soon most non-metallic deBVSs were recognised to be invisible in MRI, which allows direct assessment of vessel patency without impairment from metallic artefacts in follow-up MR-A of treated vessels [7,8].

The preserved full interpretability of treated vascular lesions in MR-A, without exposing the patients to radiation, while keeping full diagnostic control of PAD, is of utmost interest. This is due to high TLF-rates in the BTK-region, which often require repetitive imaging of the patients. Although currently available data concerning primary patency (PP) and TLF-rates of lesions treated with a deBVS seem promising, so far, only little is known about the role of MR-A in long-term follow-up of these lesions in the BTK-region [9,10]. Therefore, triggered by suspected clinically driven TLF (CD-TLF), we used MR-A as primary imaging modality to follow the patency of a recently introduced Sirolimus-eluting resorbable Tyrocore®-made scaffold (Motiv®-stent, REVA Medical, San Diego, CA, USA). Our aims were, firstly, to prove whether MR-A is reliable to detect PP of previously treated lesions and, secondly, to what extent realistic PP-rates are achievable with this new non-metallic deBVS.

## Methods

### Patients

From our institutional database 19 consecutive patients (10 female, 9 male; age: 79 ± 8.2 years [median ± MAD]) treated with the Motiv®-stent (REVA Medical, San Diego, CA, USA), a drug eluting bioresorbable vascular scaffold (deBVS), in the popliteal or the BTK-region due

Table 1. Baseline characteristics of patients treated with the investigated scaffold.

| Patients' Baseline Characteristics | |
| --- | --- |
| **patients** | |
| treated | m: 9/ f: 10 |
| age | 79 ± 8.2 years |
| included$_{censored}$ | m: $7_8$/ f: $9_9$ |
| **vascular risk factors** | |
| nicotine abuse | 8 (42%) |
| hypertension | 15 (79%) |
| diabetes mellitus | 12 (63%) |
| hyperlipidemia | 12 (63%) |
| **medical vascular history** | |
| FR-grade IIb | 7 (37%) |
| FR-grade III | 1 (5%) |
| FR-grade IV | 11 (58%) |
| preceding interventions | 11 (58%) |
| cerebrovascular disease / stroke | 10 (53%) |
| coronary heart disease | 9 (47%) |
| **target lesions** | |
| TASC II*—type A | 11 (58%) |
| TASC II*—type B | 4 (21%) |
| TASC II*—type C | 4 (21%) |
| popliteal artery– P3-segment | 1 (5%) |
| tibio-fibular trunk | 4 (21%) |
| anterior tibial artery | 10 (53%) |
| posterior tibial artery | 3 (16%) |
| fibular artery | 1 (5%) |

(values given as: Absolute number (percent) or $\bar{x}$ ± MAD; *modified TASC-II BTK-criteria [12]).

to PAD of clinical grades ≥ IIb (chronic limb threatening ischemia (CLTI) defined as grades III & IV) according to the Fontaine-Ratschow (FR) classification were identified retrospectively (see Table 1 for a full summary of baseline characteristics). Patients included in the analysis required a three dimensional contrast enhanced (3D-CE) MR-A of the lower limbs, performed after and optionally before the index treatment, which could be used for quantitative assessment of the stent patency [11]. Examination time points of MR-A resulted from checkups of our patients within the first 3 to 6 months after treatment, when all patients are routinely summoned for a clinical control examination and MR-A with complementary ultrasound examinations, if MR-A remains inconclusive or a TLF is suspected. In case of recurrent symptoms of PAD patients may always visit our outpatient clinic.

Owed to multimorbidity and reduced adherence of our PAD patients two patients were lost to follow-up and no further follow-up MR-A was available in these cases. Another patient was carrier of a pacemaker incompatible with MRI. As CT-A follow-ups were available in this case, we considered this patient just for censoring to more reliably estimate PP. This left 16 patients with a complete MR-A follow-up, where in two of these patients no pre-MR-A was available as they initially presented with a CT-A sufficient to indicate an endovascular therapy. Note that, with respect to the small sample size, the one patient receiving only CT-A examinations due to a MRI-incompatible pacemaker was considered only in the estimation of the long-term PP-

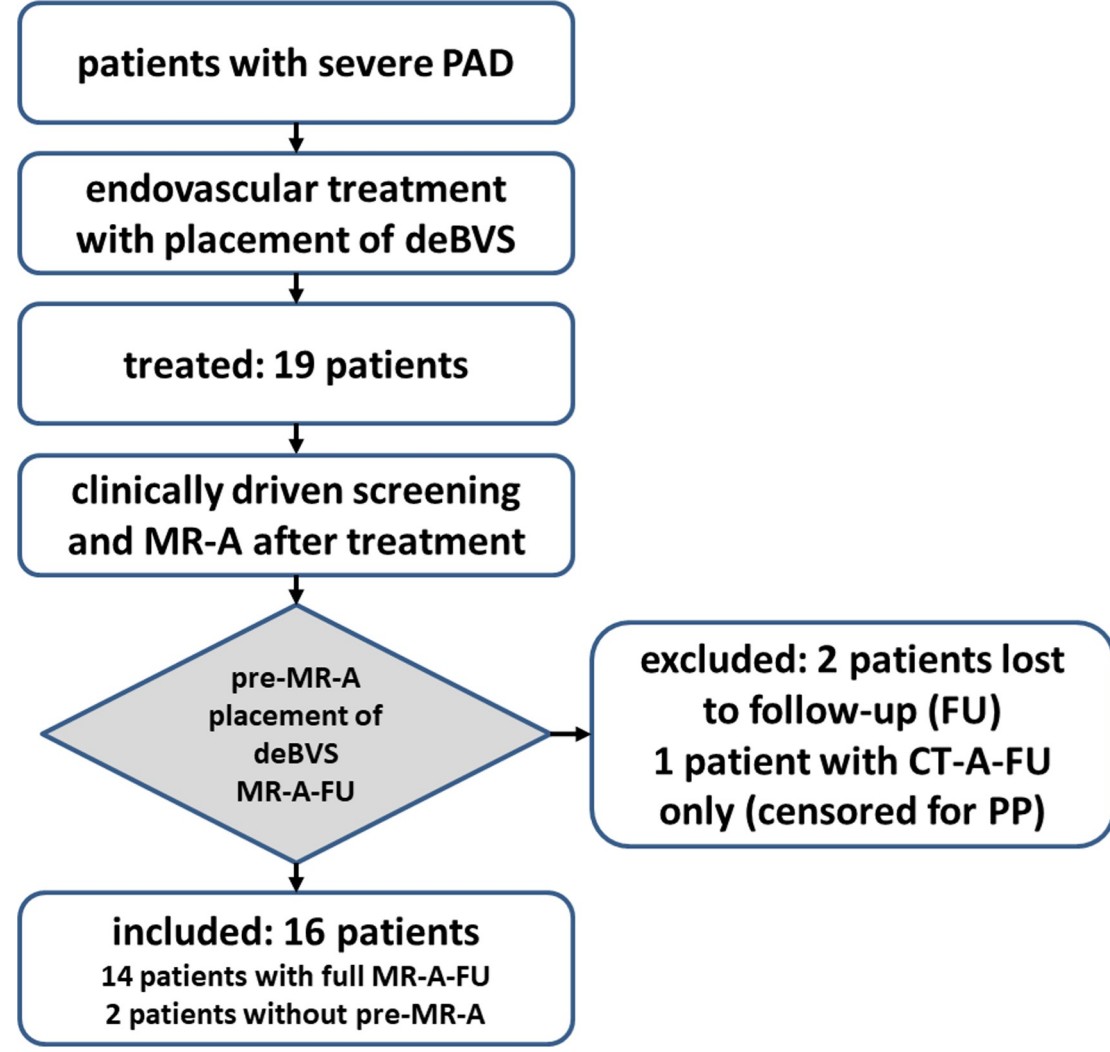

**Fig 1. Flowchart depicting the selection of the patients.** Two patients were lost to follow-up, as they did not respond to invitations to routine follow-up examinations after the interventions. One patient could only receive CT-A due to a MRI-incompatible pacemaker, but was included later in the long term primary patency (PP) analysis. Sixteen patients were eligible for MR-A assessment.

probability of the investigated deBVS. Otherwise, this patient did not contribute to any assessment in this study (Fig 1).

The study was conducted according to standards of the council of the World Medical Association and was approved by the local ethic commission of the Karl Landsteiner Privatuniversity of Health Sciences (EK Number: 1043/2023) [13]. Informed consent was waived with regard to the retrospective study design. Data collection started on 1st Nov 2023 and includes patients from 14th March 2019 to 30th June 2024. All sensible data was kept protected within the resources of the hospital in compliance with European Community (EC) regulations concerning the storage and processing of personal data. N.C., K.M. and R.K. were involved in collecting the data, where N.C. was the only person who had full access to all protected personal patients' data. After anonymisation of the records and images the assessment of the data was performed.

## Target lesions and endovascular procedure

All target lesions were rated by the institutional vascular board according to recommended modification of the Trans-Atlantic Intersociety Consensus II (TASC-II*) for the BTK-region [12]. In all patients, concomitant baseline antiplatelet therapy with aspirin $\geq$ 100 mg and clopidogrel 75 mg was started at least 5–7 days before the interventions and was prescribed for another 12 months in the follow-up period. We followed a strict 'leave nothing behind' strategy and, therefore, a deBVS was implanted in case of a flow impairing dissection or an insufficient luminal gain after percutaneous transluminal angioplasty (PTA) only [10]. The implantations were preceded by recanalisation and thoroughly preparation of the target lesion site, thereby performing PTA with either a non- or a semi-compliant balloon of size ± 0.5 mm of the measured original vessel diameter. In all patients the recently introduced Sirolimus-eluting (115 µg/device) Motiv®-stent was used, which consists of the non-metallic Tyrocore®-material that is synthesised from a polycarbonate co-polymer of tyrosine analogs and biocompatible hydroxyesters. Visibility in angiography is achieved by covalently bound Iodine atoms. Thickness of struts is 125 µm and the radial force is considered initially comparable to recently used mDES. Full strength is maintained at least for 3 to 4 months. Thereafter, the scaffold is expected to degrade completely over a period of 36 to 48 months. More details about synthesis and properties of this scaffold are described elsewhere [14–16].

The balloon-mounted scaffolds were placed centered to the target lesions and deployed at a moderate inflation speed (~ 1 bar/1 s), thereby dilating the balloon to its intended diameter. After full detachment of the deBVS, its adaption to the vessel wall and separation from the balloon were checked and the balloon carefully withdrawn. In case of an incomplete coverage of the target lesion by one single device another one was placed with minimal overlap ($\leq$ 1 mm) in direct conjunction to the first implant. Mixture of different stent types and, thus, materials, was avoided.

## MRI and 3D-CE-MR-A

3D-CE-MR-A was performed on a clinical 1.5 T scanner (Avanto Fit-SQ, Siemens Healthineers, Erlangen, Germany) using the standard body angiography coil array as provided by the manufacturer. Native T1-weighted (T1w) gradient echo imaging (TR = 3.37 ms, TE = 1.1 ms, nex = 1) with three slice stacks (88–92 slices each; nominal voxel size: $1.12 \times 1.12 \times 1.2$ mm), using a large field of view ($499 \times 499$ mm), was performed. Stacks were positioned continuously from the abdomen to the feet with minimal overlaps between them to enable seamless image-merging of angiographic full size pictures covering the vessels from the abdominal aorta to the feet. Thereafter, using dynamic bolus tracking, CE-imaging using cyclic standard contrast agents (Gadobutrol or Gadoteric acid) was performed the same way and native- and CE-images were subtracted. After merging the subtracted images to full size pictures of the lower limb vessels maximum intensity projections (MIP), depicting a 3°-step/full 180°-z-axis rotation, were computed.

## In-vitro experimental MRI

Non-metallic biodegradable scaffolds are considered to be invisible in MRI and were shown to cause no significant artefacts in coronary- or phantom-MRI examinations [7,8,17]. However, concerning the investigated Tyrocore®-deBVS no valid data about device-induced artefacts in MRI were available. Thus, an additional in-vitro MRI experiment was performed to estimate the magnitude of potential artefacts. A deBVS (3.0/27 mm MOTIV® stent) was inserted by two thirds into a bovine vessel-specimen and deployed, which left a free uncovered tail of the device outside the vessel. Then, the specimen was put inside a plastic test tube and, avoiding

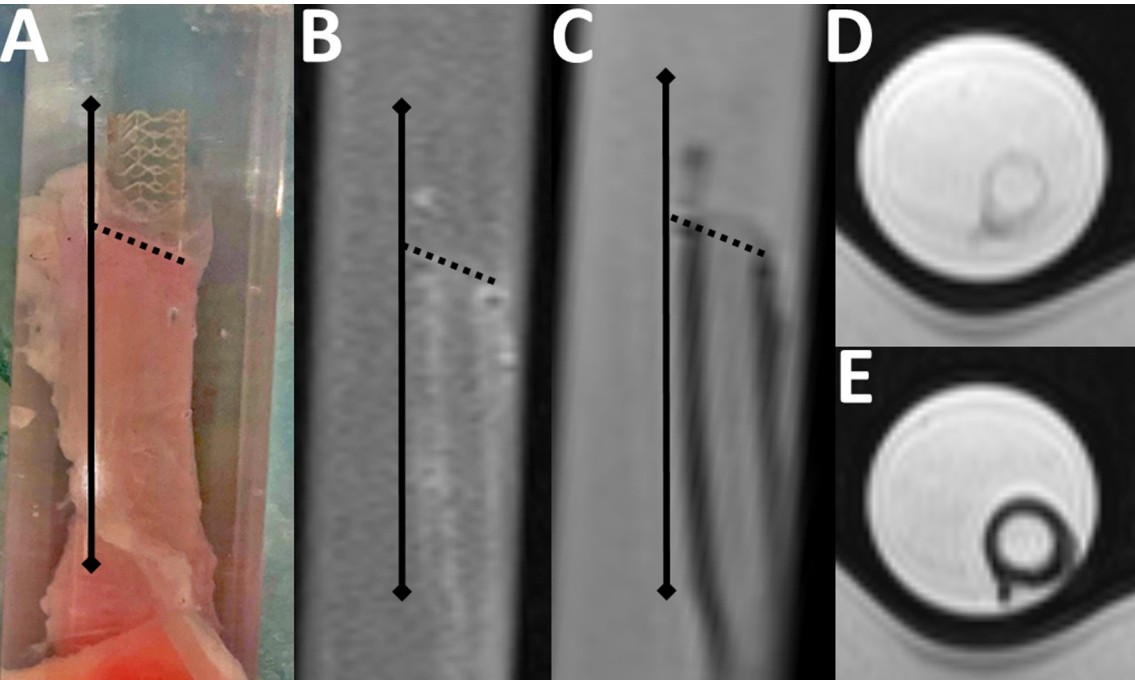

**Fig 2. In-vitro MRI of the investigated deBVS revealed no impairment of the regional signal from device-artefacts.** (A) The scaffold (stent position: Full black line) was inserted by two thirds into the carbomer-filled specimen and tube, respectively, thereby producing a stented vessel- and a free stent tail-partition (transition zone between free stent tail and stented vessel: Black dotted line). (B) The scaffold was nearly invisible in longitudinal hr-T1w-MRI, while the vessel wall exhibited a minimally increased signal. (C) The free stent tail was hardly visible in longitudinal hr-T2w-MRI, nor was the stent differentiable from the vessel wall. (D) Transverse T2w-MRI of the free stent tail and (E) the stented vessel did not show any significant degradation of MRI-signal inside or around the scaffold.

generation of air bubbles, the tube was filled with a carbomer gel, thereby also completely filling the stent- and vessel-lumen. After this, the tube was placed inside the standard 16 channel hand-coil of the clinical 1.5 T MR-system described above and high resolution (hr) T1 weighted- (T1w) and T2 weighted (T2w)- MR-imaging (nominal voxel size: 0.35 × 0.35 × 2.0 mm) was performed (Fig 2).

## Image assessment

In the first part of the investigation the stent location was exactly identified in in-vitro hr-T1w- and hr-T2w-images and the signal from ROIs covering the carbomer-filled lumen of (1) the stented specimen (vessel ROI) and (2) the free stent-tail (stent-ROI) was measured. Additionally, reference ROIs were placed directly adjacent to the intra-luminal vessel- and the stent-ROIs at the free carbomer background, which were considered to represented the maximum unbiased regional carbomer signal (reference ROIs). The amount of stent-related signal-degradation was then assessed by calculating the ratios between the signal of the stented vessel lumen and the lumen of the free stent tail vs. the related reference signal.

In the second part of the study all MR-A images were reviewed for their diagnostic quality to allow judgement of primary vessel patency before and after treatment with the deBVS. After this, all MR-A images, together with the images from the original digital subtraction angiograms (DSA) of the index treatment, were anonymised and transferred to a workstation. The image assessments were always performed in consent by two experienced readers (reader 1:

more than 10 years of radiological experience, reader 2: doctor in training for 6 months DSA and MR-A experience).

The scaffolds were localised on DSA-images and distances between well visible major side branches of the treated vessels and the scaffolds were taken to exactly localise the deBVSs on the full-size MR-A-MIPs. This, in turn, served to directly define the position of the scaffolds on the correlated MRA raw data images. ROIs were then drawn on the raw-data images covering (1) the scaffold-position (= $ROI_{deBVS}$), (2) an adjacent regular reference vessel part (= $ROI_{ref}$), and (3) the air-/background signal (= $ROI_{noise}$) near $ROI_{deBVS}$ and $ROI_{ref}$, thereby excluding spurious signal from ghosting artefacts. As the noise signal varied remarkably between the examinations the normalised intra-luminal signal (NIL-$S_{deBVS}$) of each examination, calculated from the quotient: $ROI_{deBVS}$ / $ROI_{ref}$, was used for further comparisons.

## Endpoints

In this retrospective study the primary index event was clinically-driven target lesion failure (CD-TLF), which was defined as a loss of PP in MR-A in any follow-up examination after the index procedure or as major amputation above the ankle required on the treated side or as the need for target lesion revascularisation. MR-A evaluations were always combined with a region of interest (ROI)-based quantitative assessment of the intra-luminal MR-A vessel signal. As major adverse events death or a major amputation above the ankle required on the treated side within 30 days after the index procedure were defined.

## Statistical assessment

Descriptive statistics are given as median ($\bar{x}$) and median absolute deviation (MAD) or as absolute number (n) and percentage (%).

As MR-signal ROI-data failed to conform to a normal distribution, non-parametric tests were used for group level testing. MRI-signal from in-vitro T1w- and T2w-MRI was, therefore, analysed using Kruskal-Wallis-tests (groups: vessel, stent, reference-ROI; level of significance: 0.05, post hoc analysis: Dunn tests with adjustment for multiple comparisons [Bonferroni]). In assessment of the MR-A examinations one sided Wilcoxon tests for repeated measurements with adjustment of the significance-level to p = 0.025 were used to test improvement of the vessel signal after successful deBVS-implantation. Finally, using an extended Cox model, Kaplan-Meier estimates of the deBVS related probability for PP over time were calculated [18,19].

All computations were performed using the statistical software R (version 4.1.2) invoking packages ggplot2, ggsurvfit, survival and vcd [18–22]. In-house developed R-scripts using the integrated development environment Eclipse (IDE 2020–12, Eclipse Foundation) with the StatET plug-in (v.4.4.1) were generated for automated image evaluation [23,24].

## Results

### In-vitro experiments

In in-vitro measurements absolute signal intensities in T2w-imaging were 846 ± 14 arbitrary units (a.u.) in the stented vessel lumen and 835.5 ± 19.5 a.u. in the related free carbomer background (ratio: 1.01). The free stent tail lumen signal intensity was 923.5 ± 11.5 a.u. and 935 ± 11.5 a.u., respectively, in the related carbomer background (ratio: 0.99). Comparably, signal intensities found in T1w-imaging were 259 ± 12.5 a.u. in the stented vessel lumen and 253 ± 10 a.u. in the related background (ratio: 1.03), while the signal intensity in the free stent tail lumen was 272 ± 8 a.u. and 271 ± 10 a.u., respectively, in the related carbomer background (ratio: 1.0). Neither T2w- nor T1w-experiments revealed a significant difference between the

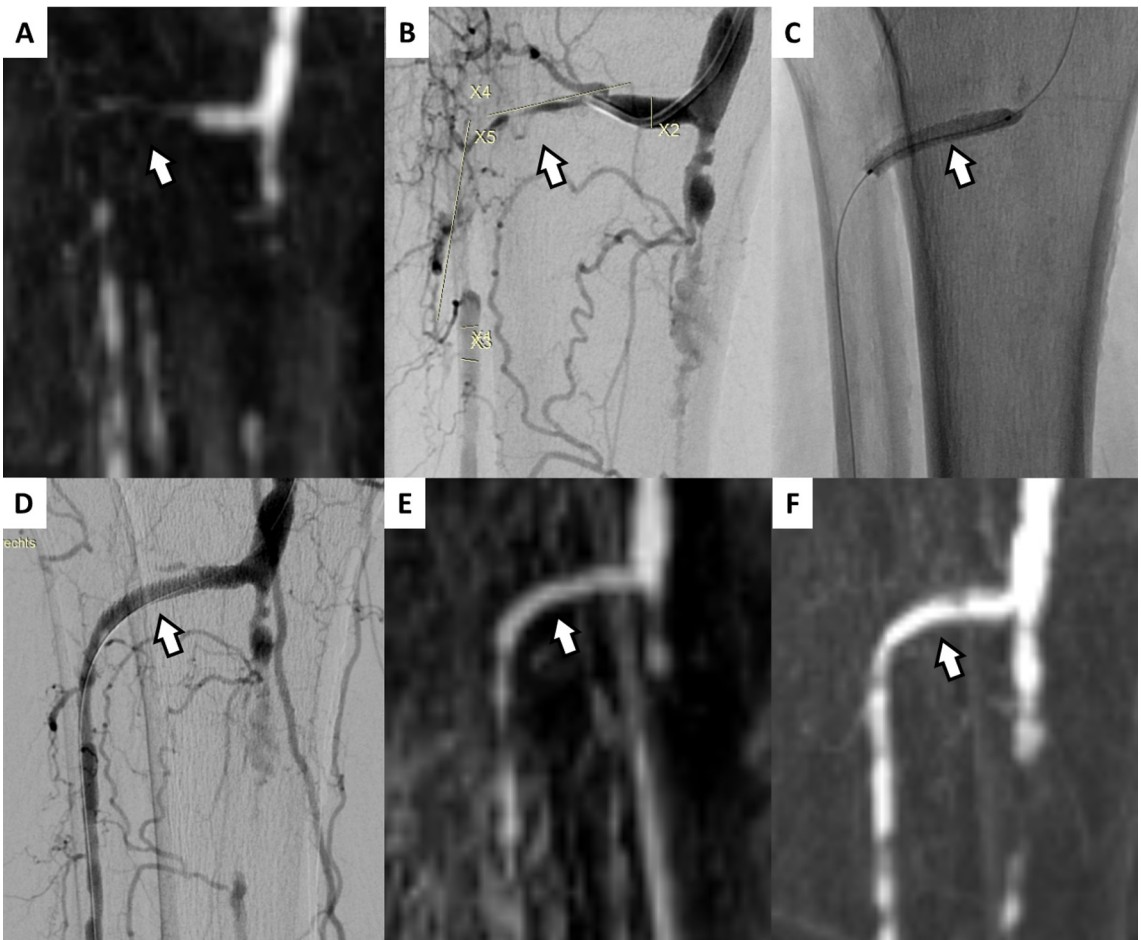

**Fig 3. Example of a patient suffering PAD of Fontaine grade IV of the right lower limb.** (A) The pre-interventional maximum intensity projection (MIP) MR-A displays an occlusion of the proximal right anterior tibial artery. (B) The initial angiogram confirmed an occlusion of this vessel bridged with numerous collaterals. (C) After recanalisation and vessel preparation a deBVS was implanted (stent site marked with white arrows). (D) The final angiogram showed sufficient reperfusion of the otherwise severely sclerotically altered target vessel. (E) MR-A follow-up performed after 27 days and (F) after 4 years (1454 days) due to recurrent PAD of Fontaine grade IV of the contralateral left lower limb. There was no impairment of the signal by the deBVS in neither MR-A, both showing a fully patent lumen inside the originally treated right target lesion.

lumen of the stented vessel-, the free stent tail and the related background (Kruskal-Wallis tests; post hoc: Dunn tests [corr.: Bonferroni]; groups: vessel, stent, reference; T2w: p> 0.259 n.s., T1w: p> 0.747 n.s.), as the scaffolds did not interfere with any of the measurements (Fig 2).

## Clinical observation

In total 16 patients with MR-A follow-up could be assessed. In clinical follow-up spontaneous in-stent thrombosis occurred before the 1st routine follow-up MR-A in two cases (13% [n = 16]) at days 37 and 91 after the index procedure. The death of two patients unrelated to the vascular interventions was perceived during the follow-up period, both after app. 19 months (subdural bleeding after trauma: n = 1, severe ischemic stroke: n = 1). Minor amputations had to be performed in 5 cases (31% [n = 16]), no major amputations or procedure related deaths were acknowledged.

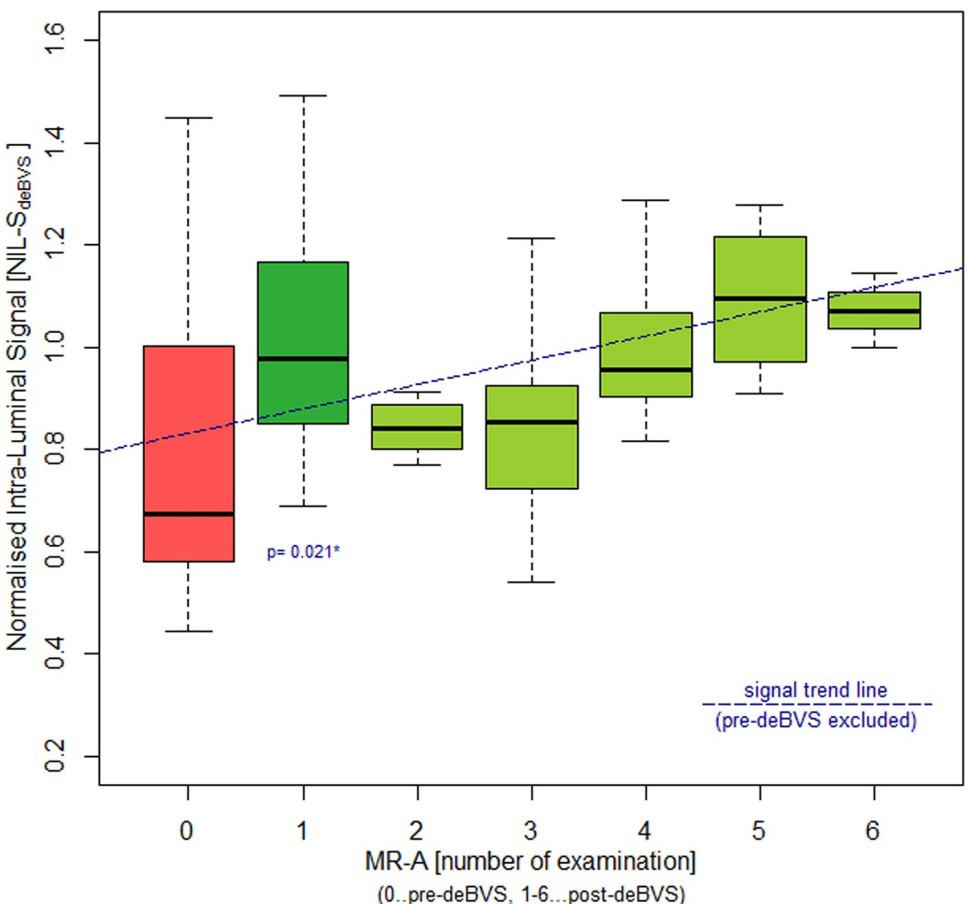

**Fig 4. Normalised intra-luminal signal (NIL-S$_{deBVS}$) of exclusively MR-A documented patients (table 2: B/A-patients): NIL-S$_{deBVS}$ increased significantly after deBVS-implantation (red bar: Pre-interventional signal, dark green bar: Signal after deBVS-implantation at 1$^{st}$ follow-up).** Successfully treated vessels exhibited a reproducible high NIL-S$_{deBVS}$ in the various follow-up examinations (light green bars), though a considerable variance with an unspecific trend of the signal to increase over time was observed.

Comparable to the in-vitro experiments, the scaffolds were found not to interfere at all with the regional vessel signal at the treated sites in MR-A (Fig 3). The cumulative assessment of all available MR-A examinations revealed a higher NIL-S$_{deBVS}$ in MR-A after successful treatment (cumulative NIL-S$_{deBVS}$: 0.64 ± 0.17 [before] vs 0.91 ± 0.18 [all follow-ups after successful treatment]), though there was a considerable variation of NIL-S$_{deBVS}$ over time. Accordingly, a direct comparison of NIL-S$_{deBVS}$ in successfully treated vessels before (0.69 ± 0.2 NIL-S$_{deBVS}$) and after (1$^{st}$ follow-up: 0.98 ± 0.28 NIL-S$_{deBVS}$) implantation of the non-metallic deBVS in 13 patients (excluded: deBVS-occlusion [n = 1]; deBVS-occlusion and no pre MRT-A [n = 1]; no pre MR-A [n = 1]) confirmed a significant increase of the intra-luminal signal in MR-A (Wilcoxon test, one sided paired test, p = 0.021 sig.). Analysis of NIL-S$_{deBVS}$ over time in these patients showed also an unspecific slight trend to increase (robust linear regression: last trimmed squares method, adjusted R$^2$: 0.154) (Fig 4). A summary of all NIL-S$_{deBVS}$-measurements is given in Table 2.

Assessment of all available examinations revealed a probability for PP of p = 0.86 (CI95: [0.71,1.0]) of the investigated deBVS after 5 years according to the Kaplan-Meier estimates.

**Table 2. Cumulative normalised deBVS-signal (NIL-S$_{deBVS}$ MR-A over time: NIL-S$_{deBVS}$ increased in target lesions in follow-up MR-A after successful treatment.**

| MR-A examination | NIL-S$_{deBVS}$ | time to index treatment [d] |
|---|---|---|
| before treatment | 0.64 ± 0.17 (n = 16) | -47 ± 36 |
| B/A-patients | 0.67 ± 0.17 (n = 14) | -44 ± 33 |
| 1st follow-up | *0.93 ± 0.27 (n = 14) | 80 ± 73 |
| B/A-patients | 0.98 ± 0.28 (n = 13) | 69 ± 53 |
| 2nd follow-up | 0.87 ± 0.09 (n = 9) | 160 ± 61 |
| B/A-patients | 0.84 ± 0.06 (n = 8) | 155 ± 48 |
| 3rd follow-up | 0.85 ± 0.15 (n = 7) | 312 ± 111 |
| B/A-patients | 0.85 ± 0.15 (n = 7) | 312 ± 111 |
| 4th follow-up | 0.95 ± 0.17 (n = 5) | 692 ± 311 |
| B/A-patients | 0.95 ± 0.17 (n = 5) | 692 ± 311 |
| 5th follow-up | 1.10 ± 0.18 (n = 4) | 1006 ± 206 |
| B/A-patients | 1.10 ± 0.18 (n = 4) | 1006 ± 206 |
| 6th follow-up | 1.07 ± 0.10 (n = 3) | 1482 ± 42 |
| B/A-patients | 1.07 ± 0.10 (n = 3) | 1482 ± 42 |

The direct comparison of NIL-S$_{deBVS}$ before and after deBVS-placement in 13 successfully treated patients revealed a significant increase (B/A-patients: *p = 0.021). Note that time points of follow-up MR-A were driven clinically and NIL-S$_{deBVS}$-values in follow-up were calculated after exclusion of stent-occlusions.

Note that for estimation of the long-term PP-probability also one patient with CT-A examinations only was added. Male and female patients generally showed a similar behaviour, despite the fact that stent occlusions occurred rather early, within the first three months, and only in female patients (Fig 5).

## Discussion

3D-CE-MR-A with clinically driven follow-up of PAD reliably depicted the patency of vessels in the BTK-region treated with a novel, recently introduced deBVS. A 5-years PP-probability of 0.87 of the investigated deBVS perfectly matched prospectively acquired 1-year mDES-data and results from alternative resorbable scaffolds [1,9]. Expectedly, the investigated non-metallic bioresorbable Sirolimus-eluting scaffold did not impair diagnostic MRI, which allowed clinically driven repetitive imaging of the BTK-vessels without exposing the patients to radiation, while artefacts commonly caused by the implantation of a mDES would have prohibited any further assessment of vessel patency [7,8,17]. To our best of knowledge, this is the first systematic analysis surveying PP of the Motiv®-deBVS (REVA Medical, San Diego, CA, USA) using MR-A in the BTK-region over a 5-years follow-up period.

Diagnostic imaging of PAD, especially in the BTK-region, is challenging due to the often impaired differentiation between arteries and overlaying veins, delays of the contrast bolus, rather small vessel diameters, vessel wall calcifications and increased patient movements. Currently, CT-A, DUS, digital subtraction angiography (DSA) and MR-A hold a similar strength of recommendation (class I, level B) concerning imaging of manifest PAD in the BTK-region, where this recommendation becomes even weaker if PAD is only suspected (class 2b, level C) [25]. DUS is most cost-effective, does not expose to ionising radiation and offers valuable hemodynamic information about vessel patency and perfusion, but is limited depicting the full extent of PAD of the lower limbs [26]. CT-A clearly displays the state of all vessels of the lower limbs and is often used to plan interventional treatment, but patients get exposed to notable doses of radiation, particularly, when repetitive diagnostic imaging is required [27,28]. Diagnostic validity of CT-A and DUS becomes reduced substantially in case of heavy vessel

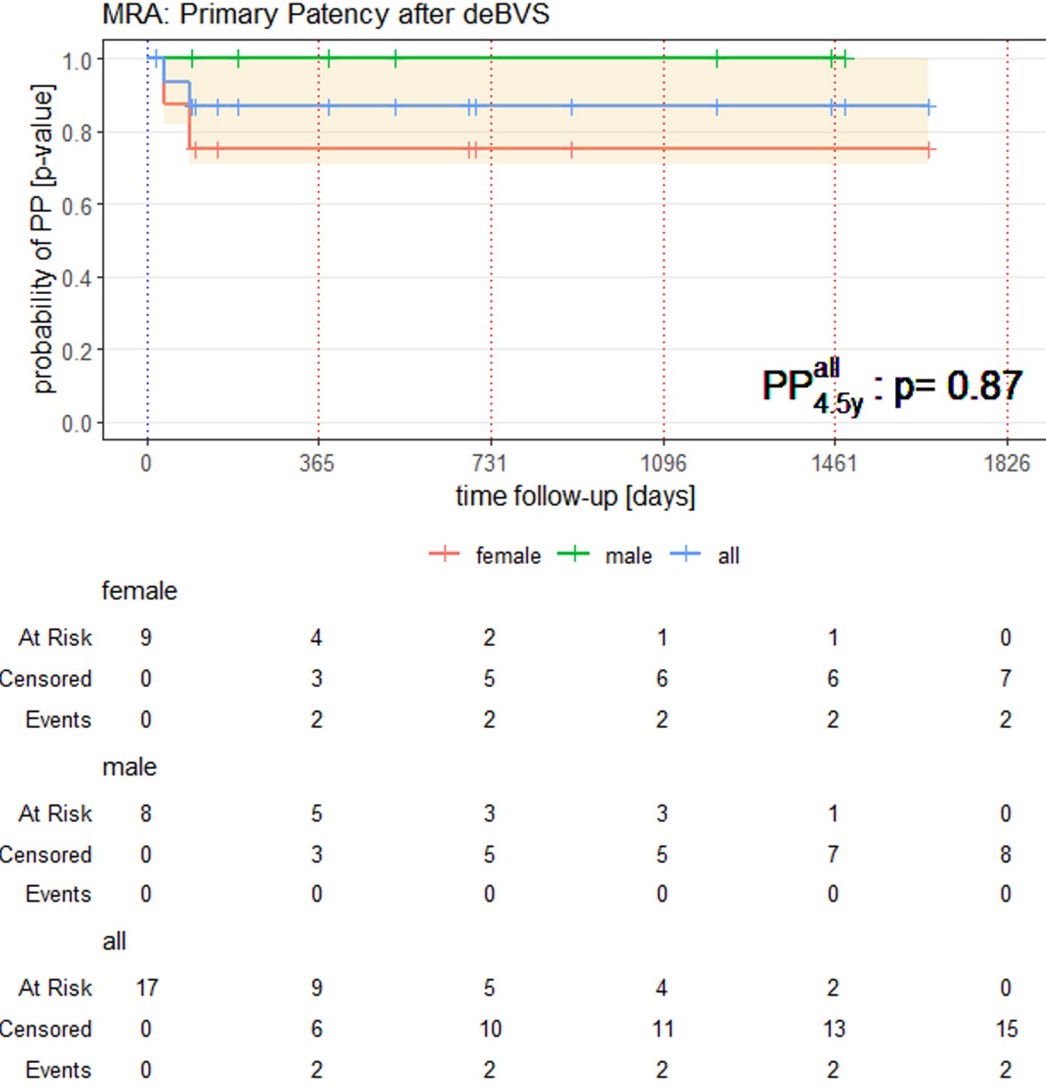

**Fig 5. Kaplan-Meier diagram estimating the probability of primary patency (PP) of the investigated deBVS followed over a 5-years period in 17 patients (CI95: Orange area).** (Please note: 16 patients exclusively received MR-A follow-up; one patient had CT-A-follow up examinations due to a MR-incompatible pacemaker). A rather high aPP of 87% after 5 years is remarkable, but may be owed to the clinically driven post-interventional screening that may lead to an overestimation of the true PP.

calcifications, which often occurs in PAD in the BTK-region. DSA offers optimal contrasts without relevant affection from vessel calcifications and is still the accepted gold standard for imaging of PAD of the lower limb arteries. However, DSA is invasive and, like CT-A, requires the administration of nephrotoxic contrast agents and exposes to ionising radiation. Moreover, the individual cumulative dose could increase considerably if CT-A is combined with often necessary repetitive endovascular revascularisations performed using DSA [29,30]. MRA, on the other hand, is neither invasive, nor is it affected by calcifications and patients do not get exposed to ionising radiation [31]. Nevertheless, diagnostic imaging of PAD in case of suspected TLF using MR-A becomes significantly impaired after implantation of a mDES that distorts the regional signal and, thus, impedes a reliable direct assessment of PP of the treated lesions in the BTK-region. A 'leave nothing behind'-strategy, adopted by many

interventionalists due to rather unsatisfying results from stenting in the BTK-region, also could not solve this problem, as, for instance, in case of persisting flow-restricting dissections or insufficient lumen gain after PTA, the implantation of a mDES is not always avoidable [10].

In the BTK-region the implantation of a mDES yielded better PP- and TLF-results than bare metal stents or standard PTA, which was primarily attributed to their drug delivery [1,9,10]. The investigated deBVS releases 60% of its Sirolimus-load to the treated vessel wall within the 1$^{st}$ month after implantation. Its radial force is specified as comparable to a mDES, but requires a strut width of 125 μm, which is clearly larger than that of a standard mDES with approximately 80 μm, but definitely smaller than that of other currently routinely used resorbable scaffolds with strut sizes around 156 μm [10,32]. While in treatment of cardio-vascular lesions strut width seems to contribute significantly to long-term PP and CD-TLF-results, in the BTK-region our 5-years PP-rate of 0.87 surveyed with MR-A excellently matches 1-year PP-rates of 0.84 and 0.85 observed with DSA for alternative resorbable scaffolds and mDESs, respectively [1,9,33,34]. In this context our findings are even more interesting, as 63% of the patients in this study presented with severe PAD of FR-grades III and IV. Anyway, despite the retrospective acquisition of the data with apparently higher losses to follow-up and a lower confidence limit of 0.71 only, our results suggest that the investigated deBVS may perform at least comparably to a mDES or an alternative resorbable scaffold in the BTK-region. These findings are also in good accordance with low rates of binary restenosis of less than 8% of another Tyrocore$^{®}$-made deBVS encountered in the coronary arteries 6 and 9 months after implantation, where cardiac MRI too remained fully interpretable [14,35].

A persistent full interpretability of vessel patency after implantation of the investigated deBVS by MRI, combined with success rates of a mDES, could represent a huge step forward in treatment of PAD in the BTK-region, all the more as this deBVS establishes an optimal pre-condition for an eventually required re-treatment in case of TLF by its full degradation 36–48 months after the implantation [1,10,36]. The unimpeded depiction and direct interpretability of vessel patency after implantation of the tested Tyrocore$^{®}$-deBVS was confirmed by calculation of the NIL-S$_{deBVS}$ in this study. Our in-vitro experiments showed the same signal ratios between the stented vessel-lumen and reference measurements as previously reported for another currently used resorbable device, which is considered invisible in MRI [7]. Neither in-vitro MRI-experiments nor MR-A examinations revealed any impairment or distortion of the regional MRI-signal (Figs 2 and 3). Accordingly, a significant increase and persistently high NIL-S$_{deBVS}$ was found in target lesions with preserved patency, while two in-stent-thromboses were safely depicted with a complete luminal signal loss in follow-up MR-A examinations after treatment. Although, with respect to radiation hygiene, this was not validated by DSA, it is conceivable that the unspecific continuous minimal increase of the NIL-S$_{deBVS}$ over time in obviously open lesions implies a positive stabilising effect on vessel patency from the investigated deBVS (Fig 4). Nevertheless, the calculation of the NIL-S$_{deBVS}$ from MR-A proved reliable to correctly indicate vessel patency in conjunction with the investigated Tyrocore$^{®}$-deBVS.

However, several limitations have to be considered. Firstly, unimpeded diagnostic imaging of the lower limb arteries despite placement of a deBVS using MR-A does not expose the patients to radiation, but requires the administration of gadolinium-contrast agents, which comprise a higher nephrotoxicity than iodine-contrast agents. This relative disadvantage is widely overcome by the clearly more physiological viscosity and the by far lower doses needed to reach a sufficient image contrast in MRI [37]. Gadolinium-contrast agents also bear a certain risk of nephrogenic systemic fibrosis that relates to reduced renal function, a serious comorbidity of PAD. When using macrocyclic contrast agents, like in this study, this risk is considered as very low [38]. Nevertheless, every administration of gadolinium must be the best

trade-off between diagnostic demands and concomitant health hazards. Secondly, the retrospective character of this study allowed a clinically driven endpoint only, which simulates an apparently high loss of patients to follow-up, since asymptomatic patients with and without true TLF were not included. However, the fact that only two patients with true CD-TLF consulted the outpatient clinic within 5 years, while all other cases were censored due to symptoms of the contralateral leg, supports the finding that the true PP-rate of this deBVS may indeed be comparably high to a mDES. Nevertheless, MR-A always provided a reliable proof of the real patency of all deBVSs. Finally, the rather small sample size prohibits any mandatory conclusions and reflects mainly our 'leave nothing behind'-strategy in the BTK-region, where a deBVS was placed only under bailout conditions. Even when implanted only under these conditions, MR-A was never impaired by the scaffold or properties of the lesions afterwards.

## Conclusion

With respect to limitations of a retrospective study and the small number of patients we found evidence that vascular lesions in PAD of FR-grades IIb-IV in the BTK-region can be treated successfully with the investigated non-metallic Tyrocore®-made deBVS with an acceptable high PP-probability. During the observed 5-years period the interpretability of the treated lesions remained fully preserved in MR-A in all cases. Due to the full degradation of this deBVS within 3–4 years after placement a full option for re-treatment of the lesions is maintained, which underlines the high importance of this proof of compatibility of this deBVS with MR-A in order to judge an eventual TLF in the BTK-region.

## Supporting information

**S1 Text. Minimal data set.** Minimal Data Set: Authors must share the "minimal data set" for their submission. PLOS defines the minimal data set to consist of the data required to replicate all study findings reported in the article.
(CSV)

**S1 Fig. The initial occlusion of the right tibiofibular trunk (shown 23 days before the index procedure) was successfully recanalised (D 0: iaDSA-image after stent placement), but was found occluded already after 37 days with persisting pain in the right calf.** The occlusion was confirmed also 141 days later. (white arrows mark the position of the deBVS).
(TIF)

## Author Contributions

**Data curation:** Christian Nasel, Mario Kirschner, Karoline Rizzi, Nicola Schweinhammer.

**Formal analysis:** Christian Nasel, Mario Kirschner.

**Investigation:** Christian Nasel, Mario Kirschner, Karoline Rizzi, Nicola Schweinhammer, Ewald Moser.

**Methodology:** Christian Nasel.

**Project administration:** Christian Nasel, Karoline Rizzi.

**Resources:** Mario Kirschner, Nicola Schweinhammer.

**Software:** Christian Nasel.

**Supervision:** Ewald Moser.

**Validation:** Karoline Rizzi, Nicola Schweinhammer, Ewald Moser.

**Visualization:** Christian Nasel.

**Writing – original draft:** Christian Nasel, Ewald Moser.

**Writing – review & editing:** Ewald Moser.

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
