## [Decision Letter · Decision Letter 0]

13 Sep 2024

PONE-D-24-33171Magnetic Resonance Angiography in Diagnostic Long-Term Follow-Up of Primary Patency of the MOTIV® Drug-Eluting Bioresorbable Vascular Scaffold in the Region Below the Knee: 5 Years of ExperiencePLOS ONE

Dear Dr. Nasel,

Thank you for submitting your manuscript to PLOS ONE. After careful consideration, we feel that it has merit but does not fully meet PLOS ONE’s publication criteria as it currently stands. Therefore, we invite you to submit a revised version of the manuscript that addresses the points raised during the review process.

We look forward to receiving your revised manuscript.

Kind regards,

Wenguo Cui, Ph.D

Academic Editor

PLOS ONE

Journal Requirements:

3. Thank you for stating the following in your Competing Interests section: NO

Please complete your Competing Interests on the online submission form to state any Competing Interests. If you have no competing interests, please state "The authors have declared that no competing interests exist.", as detailed online in our guide for authors at http://journals.plos.org/plosone/s/submit-now This information should be included in your cover letter; we will change the online submission form on your behalf.

4. We note that your Data Availability Statement is currently as follows: “All relevant data are within the manuscript and its Supporting Information files.”

Please confirm at this time whether or not your submission contains all raw data required to replicate the results of your study. Authors must share the “minimal data set” for their submission. PLOS defines the minimal data set to consist of the data required to replicate all study findings reported in the article, as well as related metadata and methods (https://journals.plos.org/plosone/s/data-availability#loc-minimal-data-set-definition). For example, authors should submit the following data: - The values behind the means, standard deviations and other measures reported; - The values used to build graphs; - The points extracted from images for analysis. Authors do not need to submit their entire data set if only a portion of the data was used in the reported study. If your submission does not contain these data, please either upload them as Supporting Information files or deposit them to a stable, public repository and provide us with the relevant URLs, DOIs, or accession numbers. For a list of recommended repositories, please see https://journals.plos.org/plosone/s/recommended-repositories. If there are ethical or legal restrictions on sharing a de-identified data set, please explain them in detail (e.g., data contain potentially sensitive information, data are owned by a third-party organization, etc.) and who has imposed them (e.g., an ethics committee). Please also provide contact information for a data access committee, ethics committee, or other institutional body to which data requests may be sent. If data are owned by a third party, please indicate how others may request data access.

Additional Editor Comments:

Suggest making revisions according to the reviewer's comments and resubmitting for peer review.

Reviewers' comments:

Reviewer's Responses to Questions

**Comments to the Author**

1. Is the manuscript technically sound, and do the data support the conclusions?

Reviewer #1: Yes

Reviewer #2: No

2. Has the statistical analysis been performed appropriately and rigorously? 

Reviewer #1: N/A

Reviewer #2: No

3. Have the authors made all data underlying the findings in their manuscript fully available?

Reviewer #1: Yes

Reviewer #2: Yes

4. Is the manuscript presented in an intelligible fashion and written in standard English?

Reviewer #1: Yes

Reviewer #2: Yes

5. Review Comments to the Author

Reviewer #1: Author used contrast enhanced MR-A of the lower limbs in 19 patients with PAD to retrospectively assess patency of a non-metallic drug eluting bioresorbable vascular scaffold (Tyrocore®) in the region below the knee. The probability of primary patency during the observation period of 5 years were computed by clinically driven MR-A censoring which triggered by an assumed target lesion failure. Additional in-vitro experiment was also prepared to prove this particular drug eluting bioresorbable vascular scaffold is compatible with MRI. The manuscript is technically sound, and I have some comments listed below:

First, are normalised intra-luminal signal (NIL-SdeBVS) and Normalised deBVS-signal (nSdeBVS) the same or they are different. If they are the same, please make it consistent across the manuscript.

Second, is it possible to include raw data to calculate table 2 "nSdeBVS" in the supporting material? And organize the data to show all the visit of each patient so that we can see the trend?

Third, since the patients number is very low could author summarize any patient who has three images of before treatment, 1st follow up and image that conclude CD-TLF in the supporting material?

Reviewer #2: 1. The sample size of the included patients is small, and the conclusions lack reliability.

2. Lines 96-99 are quite puzzling; please provide a detailed explanation.

3. In lines 102-103, to estimate long-term PP rates, only one patient who underwent CTA was included for comparison, which is not statistically valid, and conclusions cannot be drawn from this.

4. In vitro experiments are conducted; it is recommended to include in vivo animal experiments, as this would make the conclusions more convincing.

5. Lines 225-227 define MAE; however, it is not mentioned later.

6. The results in Table 2 are puzzling and do not align with the defined endpoints; it is suggested to revise Table 2 to indicate "how many individuals experienced the endpoint during each follow-up and the PP probability."

6. PLOS authors have the option to publish the peer review history of their article (what does this mean?). If published, this will include your full peer review and any attached files.

Reviewer #1: No

Reviewer #2: No

---

## [Author Response · Author response to Decision Letter 0]

30 Sep 2024

Journal Requirements:

1st point) The link proposed in Your letter is not valid (URL: https://journals.plos.org/plosone/s/file?id=wjVg/ PLOSOne_formatting_sample_main_body.pdf; accessed last: 2024.09.18). Instead we re-checked the submission guide lines (URL: https://journals.plos.org/plosone/s/submission-guidelines; accessed last: 2024.09.18). For references, as we use EndNote, the most recent style was used. We found some deviations, which were corrected. We hope that this fixes the problems during the review process.

2nd point) According to the standards published by PLoS (URL: https://journals.plos.org/plosone/s/materials-and-software-sharing; accessed last: 2024.09.18) software development was no central part of the manuscript. We employed cran R to evaluate our measurements using only standard functions approved by the publisher. All statistical functions used are publicly available with the cran R kernel or the packages already cited in the original paper.

3rd point) Sorry, we though that this was a direct question. Yes, we would like to state in the new submission: 'The authors have declared that no competing interests exist.'

4th point) Actually, all relevant data are already within the manuscript. However, a Supporting Information-file meeting the “minimal data set”-policy of PLoS was added, not least in any case this was necessary to clearly present all measurements.

5th point) The requested captions were added at the end of the manuscript.

Comments to Reviewers

Reviewer 1

1st point) We thank reviewer 1 for this comment. These abbreviations have the same meaning. During replacement of older versions, we unintentionally skipped table 2 and figure 4, when we tried to find a most convenient abbreviation for this measure. Now the abbreviation: "NIL-SdeBVS" is used throughout the paper. Figure 4 was also corrected.

2nd point) We provided a supporting information csv-sheet that comprehensively shows all data. Table 2 was rigorously reworked (see also comments to reviewer 2) and should be easier to read now. Our intention was to present the full set without any restrictions, which was obviously too confusing. However, all values of the original submission and the revision can be calculated from the supporting information csv-sheet.

3rd point) We inserted a supporting information figure demonstrating an early occlusion of the deBVS. The corresponding measurements (p19) can be derived from the supporting information csv-sheet. We hope this meets the request.

Reviewer 2

1st point) We agree that the sample size is small. Therefore, this was pointed out in the limitations. However, we disagree with reviewer 2 that the conclusions lack reliability, since the compatibility of the Tyrocore®-stent with MRI was also proven in-vitro and could be reliably confirmed with MR-A in-vivo. If this deBVS significantly annihilated the regional MR-signal, we would not have been able to judge a single patient correctly. On the contrary, we did not encounter a diagnostic restriction in any of our cases. Thus, from a statistical point of view we deal with an extremely strong positive statistical effect in favour of unrestricted MRI after implantation of the investigated deBVS. This can be tested already in a small number of cases. Though for statistical purposes a bigger number of cases would have been desirable, one cannot conclude that the direct depiction of patency of vessel segments treated with the deBVS would be not feasible. As diagnostic MRI after placement of this deBVS seems preserved, of course, we reported the number of stents censored and occluded. We do not claim that our retrospectively estimated probability of primary patency is precise enough to hold against future prospective data, but our findings provide a first and reasonable impression that this MR-A assessable deBVS may perform not worse compared to published data of metallic drug eluting stents in the BTK-region. Therefore, the conclusion that the use of a deBVS comparable to the tested one together with MR-A seems recommendable, because diagnostic and treatment options appear to be preserved, may indeed be derived even from our small sample.

2nd point) We agree that presenting the complete data in a clear and well-arranged fashion in the manuscript is difficult. A full step-by-step explanation of the collection and inclusion process of all data is now inserted at the end of the 'Patients'-subsection in the 'Methods'-chapter (revision: lines 96-109). Additionally, the flow chart demonstrating the patient collection (figure 1), was also revised and adapted to the inserted explanation. We hope that this renders the data collection easier to read.

3rd point) There was not at all a comparison between CT-A and MR-A reported in our paper. Just to get a better estimate of the PP-probability of the investigated deBVS, due to the small number of cases, we censored one patient receiving CT-A only for the Kaplan-Meyer estimates. This was clearly stated in the original submission. Nevertheless, this is now explained in more detail in the revision (revision: lines 109-113).

On the other hand, according to the strict open data policy of PLoS, we could not simply conceal the patency data of this patient, because we very well knew the state of the deBVS. This is quite different to patients lost to follow-up, where the patency state is not known at all and who, therefore, can be simply excluded. The estimated PP-probability was only used to underline the recommendation that it is worth to consider MR-A instead of diagnostic modalities exposing the patients to radiation, because these scaffolds seem to perform comparable to metallic stents but do not restrict in MR-A like the metallic ones. We do not believe that this skews the data or any of our conclusions, moreover, as this was always clearly communicated to the reader. We hope that this is also acceptable.

4th point) There is absolutely no rationale to sacrifice the life of even a single animal to convince anybody that MR-A can be used with the investigated deBVS, since, as shown in our paper, we have enough evidence in humans already and experimental data from a specimen. We are bewildered that PLoS, given the data of our paper, can agree with this suggestion, which clearly contradicts the 3-R-principle of animal experiments. Of course, taking this into account, an ethics committee in Europe would not accept such experiments and, thus, no animal experiments can or will be conducted.

5th point) We deleted this and a few other abbreviations, which did not conform to PLoS-format standards.

6th point) We recognise that table 2 may be confusing and, therefore, it was rigorously revised. We hope the actual one is easier to read now.

---

## [Decision Letter · Decision Letter 1]

30 Oct 2024

Magnetic Resonance Angiography in Diagnostic Long-Term Follow-Up of Primary Patency of the MOTIV® Drug-Eluting Bioresorbable Vascular Scaffold in the Region Below the Knee: 5 Years of Experience

PONE-D-24-33171R1

Dear Dr. Nasel,

We’re pleased to inform you that your manuscript has been judged scientifically suitable for publication and will be formally accepted for publication once it meets all outstanding technical requirements.

Kind regards,

Wenguo Cui, Ph.D

Academic Editor

PLOS ONE

Additional Editor Comments (optional):

Reviewers' comments:

Reviewer's Responses to Questions

**Comments to the Author**

1. If the authors have adequately addressed your comments raised in a previous round of review and you feel that this manuscript is now acceptable for publication, you may indicate that here to bypass the “Comments to the Author” section, enter your conflict of interest statement in the “Confidential to Editor” section, and submit your "Accept" recommendation.

Reviewer #1: All comments have been addressed

2. Is the manuscript technically sound, and do the data support the conclusions?

Reviewer #1: Yes

3. Has the statistical analysis been performed appropriately and rigorously? 

Reviewer #1: N/A

4. Have the authors made all data underlying the findings in their manuscript fully available?

Reviewer #1: Yes

5. Is the manuscript presented in an intelligible fashion and written in standard English?

Reviewer #1: Yes

6. Review Comments to the Author

Reviewer #1: Thank you for respond to my comments. Since sample size is very small. It will be great to visualize the progress of the Cumulative normalised deBVS-signal (NIL-SdeBVS) change in MR-A over time on same patient. But it seems this information is still missing.

7. PLOS authors have the option to publish the peer review history of their article (what does this mean?). If published, this will include your full peer review and any attached files.

Reviewer #1: No

---

## [Editor Report · Acceptance letter]

13 Nov 2024

PONE-D-24-33171R1 

PLOS ONE

Dear Dr. Nasel, 

I'm pleased to inform you that your manuscript has been deemed suitable for publication in PLOS ONE. Congratulations! Your manuscript is now being handed over to our production team.

Kind regards, 

on behalf of

Professor Wenguo Cui 

Academic Editor

PLOS ONE